# Does Cytokine-Release Syndrome Induced by CAR T-Cell Treatment Have an Impact on the Pharmacokinetics of Meropenem and Piperacillin/Tazobactam in Patients with Hematological Malignancies? Findings from an Observational Case-Control Study

**DOI:** 10.3390/pharmaceutics15031022

**Published:** 2023-03-22

**Authors:** Chun Liu, Pier Giorgio Cojutti, Maddalena Giannella, Marcello Roberto, Beatrice Casadei, Gianluca Cristiano, Cristina Papayannidis, Nicola Vianelli, Pier Luigi Zinzani, Pierluigi Viale, Francesca Bonifazi, Federico Pea

**Affiliations:** 1Department of Medical and Surgical Sciences, Alma Mater Studiorum-University of Bologna, 40138 Bologna, Italy; 2Clinical Pharmacology Unit, IRCCS Azienda Ospedaliero-Universitaria di Bologna, 40138 Bologna, Italy; 3Infectious Diseases Unit, Department for Integrated Infectious Risk Management, IRCCS Azienda Ospedaliero-Universitaria di Bologna, 40138 Bologna, Italy; 4Istituto di Ematologia “Seràgnoli”, IRCCS Azienda Ospedaliero-Universitaria di Bologna, 40138 Bologna, Italy

**Keywords:** CAR T-cell therapy, meropenem, piperacillin/tazobactam, cytokine release syndrome, therapeutic drug monitoring

## Abstract

Chimeric antigen receptor (CAR) T-cell therapy is a promising approach for some relapse/refractory hematological B-cell malignancies; however, in most patients, cytokine release syndrome (CRS) may occur. CRS is associated with acute kidney injury (AKI) that may affect the pharmacokinetics of some beta-lactams. The aim of this study was to assess whether the pharmacokinetics of meropenem and piperacillin may be affected by CAR T-cell treatment. The study included CAR T-cell treated patients (cases) and oncohematological patients (controls), who were administered 24-h continuous infusion (CI) meropenem or piperacillin/tazobactam, optimized by therapeutic drug monitoring, over a 2-year period. Patient data were retrospectively retrieved and matched on a 1:2 ratio. Beta-lactam clearance (CL) was calculated as CL = daily dose/infusion rate. A total of 38 cases (of whom 14 and 24 were treated with meropenem and piperacillin/tazobactam, respectively) was matched with 76 controls. CRS occurred in 85.7% (12/14) and 95.8% (23/24) of patients treated with meropenem and piperacillin/tazobactam, respectively. CRS-induced AKI was observed in only 1 patient. CL did not differ between cases and controls for both meropenem (11.1 vs. 11.7 L/h, *p* = 0.835) and piperacillin (14.0 vs. 10.4 L/h, *p* = 0.074). Our findings suggest that 24-h CI meropenem and piperacillin dosages should not be reduced a priori in CAR T-cell patients experiencing CRS.

## 1. Introduction

Chimeric antigen receptor (CAR) T-cell therapies have remarkably changed the treatment of some relapsed or refractory hematological B-cell malignancies [1]. This complex immunotherapy consists of infusing the patient’s own T-cells after previously genetically modifying these to express CAR for targeting tumor cells. Approved indications include relapse/refractory acute lymphoblastic leukemia, some B-cell lymphomas (diffuse large B-cell, primary mediastinal, high-grade, follicular, or mantle cell lymphoma) and, ultimately, multiple myeloma [2,3,4].

CAR T-cell treatment may be associated with cytokine release syndrome (CRS), an adverse effect that may occur in approximately 80% of patients [5]. CRS is characterized by high fever, hypotension, hypoxia, and ongoing injury that mimics sepsis. It usually appears within 14–21 days from CAR T-cell infusion [5,6]. Other complications occurring in the post-infusion period may be immune effector cell-associated neurologic syndrome (ICANS) and acute kidney injury (AKI) following vasodilatory shock [5,7]. 

CAR T-cell patients are at high risk of infection due to cytokine-mediated cytopenias, myelosuppression related to chemotherapy, and CRS treatment with high-dose corticosteroids and/or IL-6 inhibitors, such as tocilizumab [8]. The prevalence of infection in CAR T-cell patients may be 27–36%, with bacteremia, pneumonia, and skin and soft tissue infections being the most prevalent [8]. 

In general, all patients with hematological malignancies with febrile neutropenia, including CART T-cell patients, are at high risk of bacterial infection complications. Bloodstream infections caused by Gram-negative pathogens have a prevalence rate of 11–38% in these populations [9]. Among the most common Gram-negative pathogens are *Klebsiella pneumoniae*, *Escherichia coli*, *Enterobacter cloacae*, and *Pseudomonas aeruginosa*. The management of infections caused by these pathogens is challenged by the reduced antimicrobial susceptibility to beta-lactams in these patients. In particular, in oncohematologic patients, the susceptibility rate of piperacillin/tazobactam and meropenem has been reported to be 79.1% and 63.1%, respectively [10]. 

Current guidelines recommend an antipseudomonal beta-lactam, such as piperacillin/tazobactam, cefepime, or ceftazidime as the first-line choice for the empirical treatment of FN in patients with hematologic malignancies. These beta-lactams are preferred in clinically stable FN patients who have not had previous infections and/or colonization caused by multi-drug resistant (MDR) Gram-negative bacteria. In the absence of a positive clinical response within 2–3 days, escalation to meropenem is suggested [11,12]. 

Beta-lactams have a time-dependent pharmacodynamic activity, whose efficacy is related to the percentage of time that the free plasma concentrations are maintained above the minimum inhibitory concentration (MIC) of the bacterial pathogen (%*f*T > _MIC_) during the dosing interval. Pre-clinical data indicate that the required threshold to achieve bactericidal activity with beta-lactams is 40–70%*f*T > _MIC_ [13]. However, clinical evidence suggests that more aggressive pharmacokinetic/pharmacodynamic targets, namely, 100%*f*T > _4–6×MIC_, should be adopted to ensure better outcomes in clinical contexts characterized by high inter-individual variability, such as critically ill patients [14,15]. 

Administration of beta-lactams by 24-h continuous infusion (CI) maximizes the attainment of such a high PK/PD threshold during the entire dosing interval. Moreover, optimizing beta-lactam exposure by means of real-time therapeutic drug monitoring (TDM) has been proven effective in improving treatment outcomes with beta-lactams [13,14]. 

For the treatment of severe infections, high-dosing regimens of meropenem administered by 24-h CI has been advocated in different clinical settings [16,17]. Specifically, in order to maximize empirical treatment of Enterobacterales and *Pseudomonas aeruginosa* in FN patients with hematologic malignancies, Monte Carlo simulations suggest the use of meropenem dosages ranging from 3 to 5 g daily by 24-h CI in relation to patient renal function. 

Considering that up to 30% of CAR T-cell patients may develop AKI [7] and that beta-lactams are eliminated mainly by the renal route, it might be expected that the PK of these agents may be altered in oncohematological patients undergoing CAR-T compared with those who are not. 

The aim of this case-control study was to assess whether the pharmacokinetics of meropenem and/or piperacillin/tazobactam administered by continuous infusion were changed in oncohematological patients who received anti-CD19 CAR T-cell therapy compared with those who did not. 

## 2. Materials and Methods

### 2.1. Study Design

This evaluation retrospectively included CAR T-cell patients (case group) and oncohematological patients (control group) who underwent therapeutic drug monitoring (TDM)-guided adaptive dosage of continuous infusion meropenem and/or piperpacillin/tazobactam for the empirical treatment of febrile neutropenia. The evaluation was conducted at the IRCCS Azienda Ospedaliero-Universitaria di Bologna from January 2020 to January 2023. The ratio of the case group vs. the control group was set at 1:2 for statistical empowerment. 

All patients were treated with beta-lactam monotherapy at a standard initial dose (meropenem: 2 g loading over 1 h followed by 1 g q6h over 6 h [namely, 4 g/daily by CI]; piperacillin/tazobactam: 8/1 g loading over 1 h followed by 16/2 g over 24 h by CI). 

TDM was performed after at least 48 h from starting therapy, by assessing the meropenem or piperacillin plasma steady-state concentration (C_ss_). At our centre, all oncohematological patients who received an antipseudomonal agent underwent a program of dosing optimization that included the assessment of drug concentration, along with a clinical pharmacological interpretation of the results. This program is conducted from Monday to Friday, as described elsewhere [18]. Briefly, in the pre-analytical phase, clinicians fill in an electronic form with patient demographics (age, weight, and height), patient clinical data (diagnosis and co-medications), and drug-related information (date of starting therapy, current dosing regimen, and time of the last dose). Blood samples are collected shortly before drug administration to assess trough concentration or at any time during infusion for drugs administered by 24-h CI. After collection, the blood samples are immediately sent to the lab where they are analyzed within 1–3 h from sample delivery. The TDM results are then published on the hospital intranet early in the afternoon. The post-analytical phase starts once the TDM results are made available. Each patient request is managed by a clinical pharmacologist. Written expert clinical pharmacological advice for dose adjustments is then published on the hospital intranet before 5 p.m. 

Dosing adaptation seeks to obtain maximal effectiveness of the empirical treatment of febrile neutropenia. Therefore, a desired pharmacodynamic target of 100%t > _4–8×MIC_ [19,20] was set for all susceptible pathogens. This was achieved by considering as the MIC value the EUCAST clinical breakpoint of meropenem and piperacillin/tazobactam against *Pseudomonas aeruginosa* (namely, 2 and 8 mg/L, respectively) [21]. Consequently, meropenem C_ss_ was targeted at 8–16 mg/L and piperacillin C_ss_ at 32–64 mg/L.

### 2.2. Drug Analysis

Meropenem and piperacillin were both measured using the validated high-performance liquid chromatography tandem mass spectrometry (LC-MS/MS) method, as described below. 

#### 2.2.1. Sample Pre-Treatment

Blood samples were centrifuged for 10 min at 9000× *g*. An aliquot of 50 μL of patient plasma was added, together with a 1.25 μL solution of internal standard (final concentration 5 μg/mL). Liquid-liquid extraction was carried out using the MassTox^®^TDM Series A basic kit from Chromsystems Instruments & Chemicals GmbH, Munich, Germany. According to the manufacturer’s recommendation, an extraction buffer (25 μL) and a precipitation buffer (250 μL) were added to each sample. This solution was centrifuged for 10 min at 15,000× *g*, and an equal volume of the dilution buffer was added to the supernatant. Plasma samples used for the calibration curve and quality controls underwent the same procedure. Subsequently, using an autosampler vial in which 5 μL of the supernatant was transferred, a volume of 3 μL was injected into the LC-MS/MS system.

#### 2.2.2. Conditions of Liquid Chromatography and Mass Spectrometry

Chromatographic separation was conducted at 25 °C on a C18 column provided by the MassTox^®^TDM Series A basic kit from Chromsystems Instruments & Chemicals GmbH, Munich, Germany. The column was eluted with a gradient elution set at 0.5 mL/min using mobile phase A (0.1% formic acid in water) and mobile phase B (0.1% formic acid in methanol).

The Shimadzu UPLC system was coupled with the Sciex API 5500 Qtrap mass spectrometer with an electrospray ionization source set in positive ionization mode. Optimization of ionization conditions were performed by directly injecting drug solutions dissolved in a 50:50 volume mixture of mobile phases A and B. Mass spectrometer parameters were set as follows: medium for collision gas, 30 units for curtain gas, 5500 V for ionspray voltage, 500 °C for probe temperature, and 50 ms for dwell time. A multiple reaction monitoring (MRM)-based quantitation method technique was used. Specifically, analytes were monitored at two different transitions, namely, the quantifier ions for identification and the qualifier ions for confirmation. The Analyst 1.6 and Multiquant software Version 2.0, both provided by the spectrometer manufacturer, were used for chromatographic data acquisition, peak integration, and quantification.

#### 2.2.3. Calibration Curve and Quality Controls

The meropenem and piperacillin/tazobactam stock solution was prepared in MilliQ water at a concentration of 10 mg/mL. The calibration standards for meropenem were prepared at 0, 3, 25, and 85 mg/L, while those for piperacillin/tazobactam were prepared at 0, 8, 50, and 195 mg/L. The calibration ranges were based on plasma concentration usually observed using approved drug dosages in clinical practice, namely, 0–100 mg/L for meropenem and 0–200 mg/L for piperacillin. The quality controls were prepared at two concentrations, namely, a low concentration (13 and 20 mg/L for meropenem and piperacillin/tazobactam, respectively) and a high concentration (43 and 97 mg/L for meropenem and piperacillin/tazobactam, respectively).

#### 2.2.4. Chemical and Reagents

Meropenem, piperacillin/tazobactam sodium salt, and their isotopically labeled counterparts, 2H6-Meropenem and 2H5-Piperacillin/tazobactam, were purchased from Alsachim (Illkirch, France). Formic acid and methanol for LC-MS/MS mobile phases were purchased from CHROMASOLV (Thermofisher Scientific, Milan, Italy). A Milli-Q Direct system (Millipore Merck-Darmstadt, Germany) was used for LC–MS/MS grade water. Blank plasma was supplied for control purposes by the IRCCS Azienda Ospedaliero-Universitaria di Bologna (Bologna, Italy). Primary stock solutions for analytes and internal standards, obtained by dissolving the powder in water or dimethyl sulfoxide, were prepared to a final concentration of 10.0 and 1.0 mg/mL, respectively. All chemicals were stored at −80 °C.

#### 2.2.5. Accuracy, Precision and Limit of Quantification

The intra-day and inter-day precision and accuracy, expressed as the coefficient of variation (CV%) for the low- and high-quality controls, were <10% for both meropenem and piperacillin/tazobactam. The limits of quantification (LOQ) were 0.3 and 1.0 mg/L for meropenem and piperacillin/tazobactam, respectively.

### 2.3. Patient Clinical Data and Pharmacokinetic Analysis

Demographic, pharmacologic, and laboratory data were retrieved for each patient. Serum creatinine, serum albumin, C-reactive protein (C-RP), procalcitonin, interleukin-6 (IL-6), and interleukin-8 (IL-8) were collected on the days of each TDM assessment. The estimated glomerular filtration rate (eGFR) was calculated using the CKD-EPI formula [22]. Meropenem and piperacillin clearances (CL) were calculated using the following formula:$$CL\;\left(\frac{L}{h}\right)=\frac{Dose\;(mg)}{Css\;\left(\frac{mg}{L}\right)\times 24\;(h)}$$
where, C_ss_ is the meropenem or piperacillin steady state concentration.

Descriptive statistics were reported as the median and interquartile range (IQR) for continuous data and number with percentages for categorical data. The inter-individual variability of meropenem or piperacillin/tazobactam CL was assessed by calculating the coefficient of variation (CV%) of all the CL values obtained at each TDM assessment in each patient. 

The relationship between meropenem or piperacillin/tazobactam CL and eGFR was expressed using the Spearman rank correlation coefficient (ρ). Categorical variables were compared using the χ^2^ test or Fisher’s exact test, while continuous variables were compared using the Student *t*-test or the Mann–Whitney test. A *p*-value of <0.05 was required to achieve statistical significance. All statistical analysis and plotting was performed using R version 3.4.4 (R foundation for Statistical Computing, Vienna, Austria).

## 3. Results

The patient inclusion criteria in the study are reported in Figure 1. First, patients with hematological malignancies who underwent CAR T-cell during the study period (n = 80) were retrospectively identified. Of these, only those who were administered 24-h CI meropenem or piperacillin/tazobactam and whose therapy was optimized by TDM were included (n = 38). This group consisted of 14 patients treated with 24-h CI meropenem and 24 patients treated with 24-h CI piperacillin/tazobactam. These two groups were then matched at a 1:2 ratio to oncohematological patients treated with 24-h CI meropenem (n = 28) or piperacillin/tazobactam (n = 48) for FN but who did not receive CAR T-cell treatment. At the end, a total of 38 CAR T-cell patients was matched to 76 oncohematological patients. 

The CAR T-cell population included patients with relapse/refractory lymphomas (3 median lines of previous therapy) who were histologically grouped as follows: diffuse large B-cell lymphoma (n = 21), mantle cell lymphoma (n = 7), primary mediastinal B-cell lymphoma (n = 5), follicular lymphoma (n = 4), and high grade B-cell lymphoma (n = 1). The patients were admitted for the infusion of anti-CD19 CAR-T after lymphodepleting chemotherapy. The comparator cohort included patients with different oncohematological diagnoses: acute myeloid leukemia (n = 31), lymphoma (n = 29), acute lymphoblastic leukemia (n = 6), myeloproliferative neoplasm (n = 4), myelodysplastic syndrome (n = 4), and plasma-cell dyscrasia (n = 2). All these patients had undergone chemotherapy respective to the diagnosis and phase.

Table 1 reports the demographic and clinical characteristics of the case and the control patients treated with 24-h CI meropenem (n = 14 and n = 28, respectively). The CAR T-cell patients had a significantly lower eGFR compared with the control group (median eGFR of 63.5 vs. 94.5 mL/min/1.73 m^2^, *p* = 0.005). 

CRS occurred in 85.7% (12/14) of CAR T-cell patients, after a median (IQR) number of days from cell infusion of 4.0 (2.5–4.5). Meropenem was started after a median (IQR) number of days from cell infusion of 4.0 (1.25–4.75). During meropenem treatment, the median (IQR) range of IL-6 and IL-8 was 237.3 (49.2–2201.2) pg/mL and 46.0 (32.0–79.0) pg/mL, respectively. No patient developed AKI during treatment. The median meropenem CL in CAR T-cell patients was similar to that observed in oncohematological patients (11.1 vs. 11.7 L/h, respectively, *p* = 0.835), even if the inter-individual variability was quite high (CV% of 56% and 69.7%, respectively). 

At the first TDM assessment, the distribution of C_ss_ was similar among CAR T-cell and non-CAR T-cell treated patients (median C_ss_ of 13.5 vs. 10.85 mg/L, respectively, *p* = 0.858, Figure 2). A similar proportion of patients with Css outside the desired range [28.6% (4/14) vs. 21.4% (6/28), respectively, *p* = 0.707] was observed.

Table 2 reports the demographic and clinical characteristics of the case and control patients treated with 24-h CI piperacillin/tazobactam (n = 24 and n = 48, respectively). No statistically significant difference was observed in any of the parameters between the CAR T-cell and the non-CAR T-cell patients. 

CRS was observed in 95.8% (23/24) of CAR T-cell patients, after a median (IQR) number of days from cell infusion of 3.0 (2.0–4.0). Piperacillin/tazobactam was started after a median (IQR) number of days from cell infusion of 2.0 (0.0–4.00). During piperacillin/tazobactam treatment, the median (IQR) range of IL-6 and IL-8 was 69.4 (29.4–561.8) pg/mL and 56.0 (29.0–110.0) pg/mL, respectively. AKI occurred only in one CAR T-cell patient between day 3 and day 6 (median CL_CR_ value of 27 mL/min/1.73 m^2^), and renal function gradually recovered from day seven onward. In this patient, IL-6 levels and piperacillin CL were 4899 pg/mL and 3.78 L/h, respectively, during the AKI phase. Then, piperacillin CL increased up to 16.21 L/h when CL_CR_ returned to normal values (102 mL/min/1.73 m^2^). Median piperacillin CL was similar between the CAR T-cell and oncohematological patients (CL of 14.0 vs. 10.4 L/h, respectively, *p* = 0.074), but with very high inter-individual variability (CV% of 92.6% and 112.4%, respectively).

At the first TDM assessment, the distribution of C_ss_ was similar between CAR T-cell and non-CAR T-cell treated patients (median C_ss_ of 42.8 vs. 57.3 mg/L, respectively, *p* = 0.153, Figure 3). A similar proportion of patients with C_ss_ out of the desired range (25.0% [6/24] vs. 16.7% [8/48], respectively, *p* = 0.529) was observed.

A mild relationship was observed between meropenem CL and eGFR in the CAR T-cell patients (Spearman’s ρ = 0.34) but no relationship was observed in the oncohematological patients (Spearman’s ρ = 0.08). Similarly, for piperacillin, no relationship was observed between piperacillin CL and eGFR in both the CAR T-cell and oncohematological patients (Spearman’s ρ = 0.01).

## 4. Discussion

This investigation first reported the comparative PK of 24-h CI meropenem and piperacillin/tazobactam in CAR T-cell vs. non-CAR T-cell patients. Our findings suggest that the CL of both these beta-lactams should not be affected by CAR T-cell treatment. 

Indeed, almost all of the CAR T-cell patients experienced CRS, as evidenced by the high levels of both IL-6 and IL-8, and according to previously reported data [23,24,25]. CRS is an excessive and dysregulated immune response with increased secretion of pro-inflammatory cytokines, such as IL-2, IL-6, IL-10, and TNF-alfa [5]. This response is commonly associated with CAR T-cell therapy [5,26]. Among these cytokines, IL-6 appears to be a key driver of CRS. IL-6 is a pleiotropic cytokine that has been described as having both pro- and anti-inflammatory properties. IL-6 is produced directly by CAR T-cells after infusion, but it is also released by endothelial cells in response to pro-inflammatory signals, including TNF-α and hypoxia, and in response to tissue injury and organ failure. On target cells, IL-6 acts by binding to its receptor. This triggers gp130 and activates the Jak/STAT signaling pathway, which, in turn, activates STAT3 [27]. CRS was already associated with a downregulation of the CYP3A4-mediated drug biotransformation [28,29]. Additionally, CRS is closely associated with both AKI and chronic kidney disease [7,30,31]. It is likely that IL-6 plays a major role in kidney injury by causing acute tubular injury [32,33]. Pre-clinical models show that in nephrotoxin-induced AKI, IL-6 expression is enhanced more than a hundred-fold in the kidneys, mainly in the renal tubular epithelial cells; it is also strongly correlated with kidney damage [32]. However, mice with IL-6 deficiency and with reduced migration of neutrophil cells did not suffer from the consequences of kidney insult. This reinforces the role of IL-6-mediated neutrophil activation as one of the main mechanisms involved in AKI. Moreover, it has also been shown that IL-6 reduces endothelial nitric oxide production and adiponectin expression, thus suggesting the role of IL-6 also in patients with chronic kidney disease by inducing chronic vascular disease. Interestingly, a retrospective study conducted on 646 critically ill Japanese patients whose IL-6 levels were measured at ICU admission, showed that patients with higher levels of IL-6 (1189–2,346,310 pg/mL) had significantly higher in-hospital 90-day mortality, lower urine output, and increased probability of persistent anuria for ≥12 h compared with patients with lower IL-6 levels (1.5–150.2 pg/mL) [34]. 

On these bases, it could be expected that the elimination of meropenem and piperacillin/tazobactam would be affected after CAR T-cell treatment of oncohematological patients. However, this was not the case as the CAR T-cell patients had both meropenem and piperacillin CL similar to the control group, with values even higher than observed in other patient populations. Population PK studies of CI meropenem carried out in critically ill and oncohematological patients showed meropenem CL ranging from 5.3 to 13.04 L/h [16,35,36,37,38,39]. For piperacillin, four population PK studies and one observational PK study showed drug CL ranging from 5.6 to 13.8 L/h [17,40,41,42]. Overall, these observations may be explained considering that most of our patients (83/114, 72.8%) had median eGFR > 60 mL/min/1.73 m^2^ and the occurrence of transient AKI was observed in only one patient. It should be noted that this patient showed high values of IL-6 and a low value of piperacillin CL, suggesting that CRS-induced AKI may reduce the clearance of drugs eliminated by the kidneys. We are aware that the low incidence of AKI in our population may be due to the lower levels of IL-6 that we observed in our CAR T-cell patients compared with the values reported by [30]. Moreover, we also cannot exclude that the prompt administration of tocilizumab––a recombinant humanized anti-IL-6 receptor monoclonal antibody––to most of our patients may have attenuated the rise of IL-6.

With regard to achieving the desired efficacy targets, no difference was observed between the CAR T-cell and non-CAR T-cell treated patients. However, it is worth noting that up to 28.6% and 25% of patients treated with CI meropenem and piperacillin/tazobactam, respectively, did not attain the desired C_ss_ at the first TDM assessment. 

Surprisingly, the relationship between beta-lactam CL and eGFR was only mild for meropenem in the CAR T-cell patients, and absent for piperacillin in the CAR T-cell patients and for both meropenem and piperacillin in the oncohematological patients. This finding is in contrast to expectations based on previous data, which shows that eGFR is a significant covariate on both meropenem and piperacillin/tazobactam clearance in different patients populations, including oncohematological patients [16,17]. However, eGFR was recently reported to account for no more than 54% of the variability of meropenem elimination in critically ill patients [43], as this antibiotic is also eliminated by tubular secretion [44]. With regard to piperacillin clearance, several studies show saturative elimination occurs at therapeutic dosages, which makes drug exposure unpredictable and uncorrelated to eGFR [17]. In any case, it has already been documented how the pharmacokinetics of antibiotics predominantly cleared by the renal route may be greatly modified in patients with oncohematological malignancies [45,46]. Given that up to 25–30% of treated patients do not attain the desired target concentration at the first TDM assessment, and given the unreliability of eGFR in guiding dose adjustments both in CAR T-cell patients and in oncohematological patients, TDM may be beneficial for dosage adjustments in both CAR T-cell treated patients and oncohematological patients, similarly to what has been observed in critically ill patients [47]. 

Moreover, beta-lactam optimization by means of TDM may acquire special relevance for patients with augmented renal function, a condition that often occurs in oncohematologic patients [48]. In this patient population, dose adjustments should be based on measured rather than estimated renal function, as eGFR has been reported to either underestimate or overestimate measured creatinine clearance in different studies [49,50]. 

We recognize that our study has some limitations. The limited sample size and the estimation rather than measurement of creatinine CL must be acknowledged. Moreover, as only one patient experienced CRS-induced AKI in our cohort, it is difficult to draw definitive conclusions about the effect of CAR T-cell treatment on beta-lactam disposition. However, we believe that this study may be of interest to clinicians since our findings suggest that the treatment of febrile neutropenia with 24-h CI piperacillin/tazobactam or meropenem in CAR T-cell patients should be based on the same dosing regimens used for non-CAR T-cell patients.

In conclusion, our preliminary findings suggest that 24-h CI meropenem and piperacillin dosages should not be reduced a priori in CAR T-cell treated patients experiencing CRS, as CRS-induced AKI occurs rarely in CAR T-cell treated patients. However, clinicians should carefully monitor renal function in these patients, as drug accumulation may occur as soon as AKI develops. Large prospective studies are warranted to confirm these findings.

## Figures and Tables

**Figure 1 pharmaceutics-15-01022-f001:**
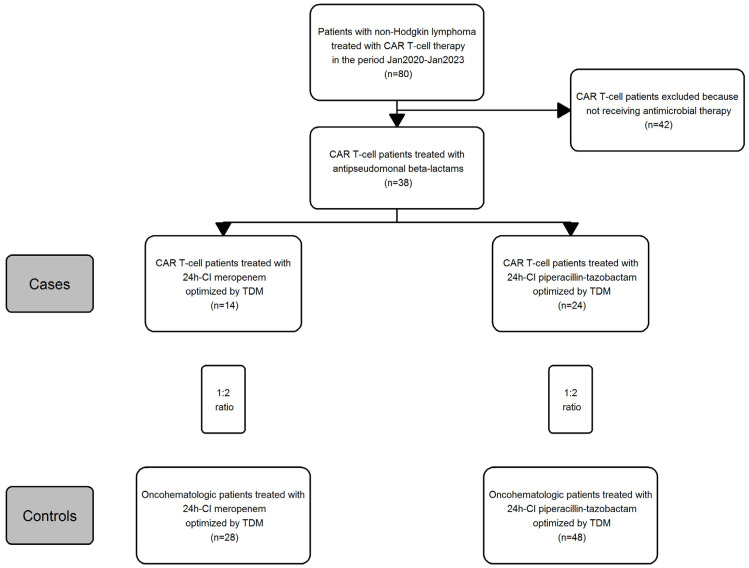
Flowchart of patient inclusion criteria in the study.

**Figure 2 pharmaceutics-15-01022-f002:**
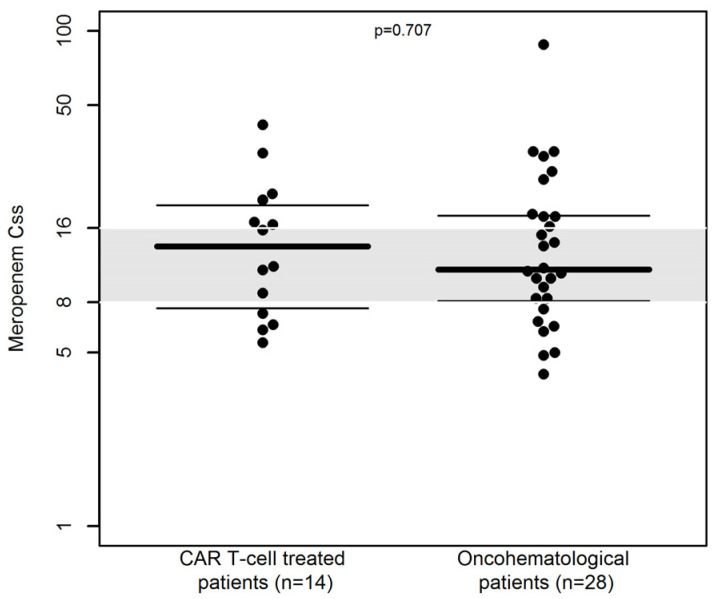
Beeswarm plot of the distribution of C_ss_ of 24-h continuous infusion meropenem (n = 42) in CAR T-cell treated patients and in oncohematological patients at first TDM assessment. The gray shaded area represents the desired therapeutic range (C_ss_ of 8–16 mg/L). The dashed line is the median. The dotted lines represent the 25th and 75th percentiles.

**Figure 3 pharmaceutics-15-01022-f003:**
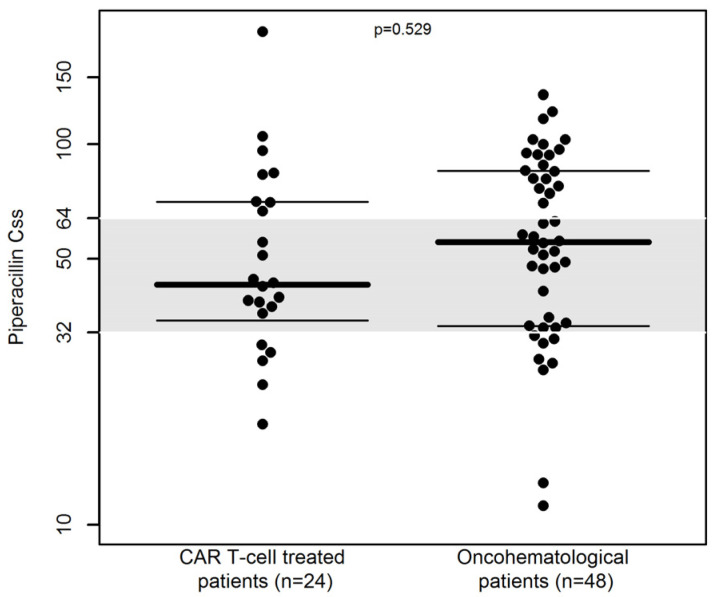
Beeswarm plot of the distribution of C_ss_ of 24-h continuous infusion piperacillin/tazobactam (n = 72) in CAR T-cell treated patients and in oncohematological patients at first TDM assessment. The gray shaded area represents the desired therapeutic range (C_ss_ of 32–64 mg/L). The dashed line is the median. The dotted lines represent the 25th and 75th percentiles.

**Table 1 pharmaceutics-15-01022-t001:** Demographic and clinical characteristics of CAR T-cell patients and oncohematological patients receiving 24-h continuous infusion meropenem (n = 42).

Variable		CAR T-Cell Treated Patients	Oncohematological Patients	*p*-Value
Number of patients (n)		14	28	-
Age (year)		61.5 (49–65)	62.5 (55–69.5)	0.371
Gender (male/female)		9/5	21/7	0.491
Weight (kg)		88.8 (71.0–97.0)	73.4 (63.0–80.0)	0.016
Height (m)		1.75 (1.64–1.80)	1.70 (1.64–1.75)	0.926
Creatinine (mg/dL)		1.08 (0.85–1.31)	0.69 (0.49–0.94)	<0.001
eGFR (mL/min/1.73 m^2^)		63.5 (54.0–90.5)	94.5 (78.8–117.3)	0.005
Reasons for meropenem				
	FN (n, %)	11 (78.7)	21 (75.0)	1.000
	BSI (n, %)	1 (7.1)	5 (17.9)	0.645
	HAP(n,%)	1 (7.1)	2 (7.1)	1.000
	UTI (n, %)	1 (7.1)	0 (0)	0.333
Meropenem treatment				
	Drug daily dose (g)	4.0 (4.0–4.0)	4.0 (4.0–4.0)	0.476
	Treatment duration (day)	9.0 (7.0–11.8)	13.0 (8.0–17.0)	0.101
	C_ss_ (mg/L)	11.0 (7.0–15.1)	12.0 (6.7–17.5)	0.852
	CL (L/h)	11.1 (7.9–21.2)	11.7 (8.2–20.1)	0.835

ALL, acute lymphocytic leukemia; AML, acute myeloid leukemia; BSI, bloodstream infection; CL, clearance; C_ss_, steady-state concentration; eGFR, estimated glomerular filtration rate; FN, febrile neutropenia; HAP, hospital-acquired pneumonia; UTI, urinary tract infection. Data are presented as median (IQR) for continuous variables and as a number (%) for categorical variables.

**Table 2 pharmaceutics-15-01022-t002:** Demographic and clinical characteristics of CAR T-cell patients and oncohematological patients receiving 24-h continuous infusion piperacillin/tazobactam (n = 72).

Variable		CAR T-Cell Treated Patients	Oncohematological Patients	*p*-Value
Number of patients (n)		24	48	-
Age (year)		61 (52–64)	64.5 (47–74)	0.187
Gender (male/female)		14/10	29/19	1.000
Weight (kg)		70 (61.5–82.3)	70.0 (60–80)	0.236
Height (m)		1.70 (1.63–1.76)	1.70 (1.64–1.76)	0.556
Creatinine (mg/dL)		0.99 (0.81–1.18)	0.81 (0.69–1.13)	0.153
eGFR (mL/min/1.73 m^2^)		76.5 (60.0–96.2)	95.5 (59.8–105.3)	0.186
Reasons for piperacillin/tazobactam				
	FN (n, %)	20 (83.3)	35 (72.9)	0.390
	BSI (n, %)	2 (8.3)	10 (20.8)	0.314
	HAP (n,%)	1 (4.2)	2 (4.2)	1.000
	UTI (n, %)	1 (4.2)	1 (2.1)	1.000
Piperacillin/tazobactam treatment				
	Drug daily dose (g)	18.0 (13.5–18.0)	18.0 (13.5–18.0)	0.522
	Treatment duration (day)	6.0 (5.0–14.0)	9.0 (7.0–12.0)	0.394
	C_ss_ (mg/L)	43.7 (34.6–65.3)	58.4 (34.8–90.1)	0.058
	CL (L/h)	14.0 (9.0–19.3)	10.40 (6.38–17.2)	0.074

ALL, acute lymphocytic leukemia; AML, acute myeloid leukemia; BSI, bloodstream infection; CL, clearance; C_ss_, steady-state concentration; eGFR, estimated glomerular filtration rate; FN, febrile neutropenia; HAP, hospital-acquired pneumonia; UTI, urinary tract infection. Data are presented as median (IQR) for continuous variables and as a number (%) for categorical variables.

## Data Availability

Not applicable.

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
