# Peer review of "Does Cytokine-Release Syndrome Induced by CAR T-Cell Treatment Have an Impact on the Pharmacokinetics of Meropenem and Piperacillin/Tazobactam in Patients with Hematological Malignancies? Findings from an Observational Case-Control Study"

_pharmaceutics, 2023, doi:10.3390/pharmaceutics15031022_

Round 1
Reviewer 1 Report
Liu et al. investigated the impact of CAR T-cell therapy on the pharmacokinetics of meropenem and piperacillin/tazobactam in oncohematological patients. The activation of immune cells and cytokine release syndrome (CRS) associated with CAR T-cell therapy may cause changes in drug clearance, particularly affecting organs such as the liver and kidneys. CRS is a common complication of CAR T-cell therapy characterized by the release of cytokines and inflammatory mediators, which can affect different organ systems, including those involved in drug clearance. This study provides evidence on the potential impact of CAR T-cell therapy on the pharmacokinetics of meropenem, piperacillin/tazobactam, and other antibiotics in oncohematological patients.
Some suggestions to improve the manuscript are as follows:
1. Consider including CRS in the title of the manuscript.
2. Since only one patient developed CRS-induced AKI, it is challenging to draw definite conclusions from this study, as most patients did not experience AKI.
3. It would be informative to report the CL of meropenem and piperacillin in the patient who developed CRS-induced AKI.
4. There are a few typos in the manuscript, such as n = 4. Please ensure there are spaces before and after the equal sign.
Author Response
Liu et al. investigated the impact of CAR T-cell therapy on the pharmacokinetics of meropenem and piperacillin/tazobactam in oncohematological patients. The activation of immune cells and cytokine release syndrome (CRS) associated with CAR T-cell therapy may cause changes in drug clearance, particularly affecting organs such as the liver and kidneys. CRS is a common complication of CAR T-cell therapy characterized by the release of cytokines and inflammatory mediators, which can affect different organ systems, including those involved in drug clearance. This study provides evidence on the potential impact of CAR T-cell therapy on the pharmacokinetics of meropenem, piperacillin/tazobactam, and other antibiotics in oncohematological patients. Some suggestions to improve the manuscript are as follows:
1. Consider including CRS in the title of the manuscript.
R.: We thank the referee for this suggestion. The title of the manuscript has been modified to include CRS. Now the title is: “Does cytokine-release syndrome induced by CAR T-cell treatment have an impact on the pharmacokinetics of meropenem and piperacillin/tazobactam in patients with hematological malignancies? Findings from an observational case-control study”.
2. Since only one patient developed CRS-induced AKI, it is challenging to draw definite conclusions from this study, as most patients did not experience AKI.
R.: We perfectly agree with this referee’s consideration. Therefore, we added a statement in the limits (end of the Discussion) on the fact that it is challenging to draw definitive conclusions as only 1 subject experienced CRS-induced AKI.
3. It would be informative to report the CL of meropenem and piperacillin in the patient who developed CRS-induced AKI.
R.: We thank the referee for this very important suggestion. Accordingly, we added in the Results the piperacillin CL value of this patient both during AKI phase and during recovery. Interestingly, piperacillin CL was low (3.78 L/h) when CLCR was 27 mL/min/1.73 m2, then it increased to 16.2 L/h when renal function returned to normal values (102 mL/min/1.73 m2). Moreover, a proper statement on the effect of CRS-induced AKI on the disposition of meropenem and piperacillin/tazobactam was also added at the end of the Discussion.
4. There are a few typos in the manuscript, such as n = 4. Please ensure there are spaces before and after the equal sign.
R.: Thanks. The entire manuscript has been revised for typos and spelling errors.
Reviewer 2 Report
This study has very limited number of study subjects enrolled and there is no proper mentioning of study design. Therefore the study has less statistical power. However this is patient based data, therefore any observation from this type of data will be scientifically important.
Were there any prospective experiments performed are this is just a retrospective data.
When (at what stage) was the patient sample (Plasma) obtained and how was it stored until downstream processing.
There is no clear description of patient inclusion and exclusion criteria so its difficult to understand the experimental workflow.
The enrolment of study subjects has been written as January 2020 to January 2023 in the text but in figure 1 it is written as January 2020 to January 2022.
One more concern that I am struggling to understand is that the title of the study and the study design. The authors have clearly stated in the title " Does CAR T-cell treatment have an impact on the pharmacokinetics of meropenem and piperacillin/tazobactam in oncohematological patients? Findings from an observational case-control study" that this study is performed on oncohematological patient data. Nonetheless, the study design shows two study groups CAR T-cell patients (case group) and oncohematological patients (control group). Why is the main study subject is enlisted as control group?
As this is based on retrospective data with prospective experiments (As per my understanding) the experiments performed in this manuscript are according to the guidelines so it seems correct. However the overall representation is poor and need some extensive editing.
Author Response
This study has very limited number of study subjects enrolled and there is no proper mentioning of study design. Therefore the study has less statistical power. However this is patient based data, therefore any observation from this type of data will be scientifically important.
R.: We thank the referee for appreciating our work
Were there any prospective experiments performed are this is just a retrospective data.
R.: This is a retrospective study. Even if patients underwent therapeutic drug monitoring of meropenem or piperacillin/tazobactam during treatment, the selection of CAR T-cell and the matching with oncohematologic patients occurred retrospectively. This concept has been rephrased and is reported in the first paragraph of the Method section.
When (at what stage) was the patient sample (plasma) obtained and how was it stored until downstream processing.
R.: Blood samples were collected after at least 48h from starting therapy, i.e. at steady state-condition, anytime during 24-h continuous infusion administration. After collection, blood samples were sent immediately to the lab, where they were analyzed in the same day, in order to provide dose-adjustment recommendation within mid-afternoon. This information was added in the third paragraph of the Methods.
There is no clear description of patient inclusion and exclusion criteria so it’s difficult to understand the experimental workflow.
R.: We apologize for not having adequately described the inclusion/exclusion criteria. First, we selected all those oncohematologic patients who underwent CAR T-cell during the study period (Jan 2020 – Jan 2023). Second, among this group, we included only those patients who received antimicrobial treatment with 24h-CI meropenem (n=14 patients) or 24h-CI piperacillin/tazobactam (n=24) optimized by TDM. Third, we retrieved from the medical records a double number of oncohematological patients who received the same beta-lactams but who did not underwent CAR T-cell therapy (n=28 patients with 24h-CI meropenem and n=48 patients with 24h-CI piperacillin/tazobactam). The first paragraph of the Result was rephrased to better explain patients’ inclusion/exclusion criteria.
The enrolment of study subjects has been written as January 2020 to January 2023 in the text but in figure 1 it is written as January 2020 to January 2022.
R.: We apologize for this error. The correct enrolment study was Jan 2020 to Jan 2023. The figure was corrected accordingly.
One more concern that I am struggling to understand is that the title of the study and the study design. The authors have clearly stated in the title "Does CAR T-cell treatment have an impact on the pharmacokinetics of meropenem and piperacillin/tazobactam in oncohematological patients? Findings from an observational case-control study" that this study is performed on oncohematological patient data. Nonetheless, the study design shows two study groups CAR T-cell patients (case group) and oncohematological patients (control group). Why is the main study subject is enlisted as control group?
R.: We thank the referee for noticing this aspect. All the patients included in this study were oncohematological patients, but some of them were treated with CAR T-cells while others were not. We agree with the referee that this may generate confusion in the title. Therefore, we substitute the term “oncohematological pateints” with “patients with hematological malignancies” in order to avoid confusion with the name of the case and control groups.
As this is based on retrospective data with prospective experiments (As per my understanding) the experiments performed in this manuscript are according to the guidelines so it seems correct. However the overall representation is poor and need some extensive editing.
R.: We thank the referee for the deep revision of our manuscript that improved the quality and the presentation of our research. In particular, we clearly specified that the nature of this study was retrospective, we added a description of patient inclusion/exclusion criteria in the Results and we corrected the date of the study period in Figure 1.
Round 2
Reviewer 1 Report
No further comment
Reviewer 2 Report
The manuscript has been revised according to the comments. I have no further queries.